# Assignment of individual structures from intermetalloid nickel gallium cluster ensembles

Maximilian Muhr[1,5], Johannes Stephan [1,5], Lena Staiger[1], Karina Hemmer[1], Max Schütz[1], Patricia Heiß[1], Christian Jandl[1], Mirza Cokoja[1], Tim Kratky [1], Sebastian Günther [1], Dominik Huber[1], Samia Kahlal[2], Jean-Yves Saillard [2], Olivier Cador[2], Augusto C. H. Da Silva[3], Juarez L. F. Da Silva[3], Janos Mink[4], Christian Gemel[1] & Roland A. Fischer [1✉]

Poorly selective mixed-metal cluster synthesis and separation yield reaction solutions of inseparable intermetalloid cluster mixtures, which are often discarded. High-resolution mass spectrometry, however, can provide precise compositional data of such product mixtures. Structure assignments can be achieved by advanced computational screening and consideration of the complete structural space. Here, we experimentally verify structure and composition of a whole cluster ensemble by combining a set of spectroscopic techniques. Our study case are the very similar nickel/gallium clusters of $M_{12}$, $M_{13}$ and $M_{14}$ core composition $Ni_{6+x}Ga_{6+y}$ ($x + y \leq 2$). The rationalization of structure, bonding and reactivity is built upon the organometallic superatom cluster $[Ni_6Ga_6](Cp^*)_6 = [Ga_6](NiCp^*)_6$ (**1**; $Cp^* = C_5Me_5$). The structural conclusions are validated by reactivity tests using carbon monoxide, which selectively binds to Ni sites, whereas (triisopropylsilyl)acetylene selectively binds to Ga sites.

[1] Department of Chemistry and Catalysis Research Center, Technical University of Munich, Lichtenbergstraße 4, D-85748 Garching, Germany. [2] Univ Rennes CNRS, ISCR-UMR 6226, F-35000 Rennes, France. [3] São Carlos Institute of Chemistry, University of São Paulo, P. O. Box 780, 13560-970 São Carlos, SP, Brazil. [4] Hungarian Academy of Sciences, Institute of Material and Environmental Chemistry, Research Centre for Natural Sciences, Magyar tudósok körútja 2, H-1117 Budapest, Hungary. [5] These authors contributed equally: Maximilian Muhr, Johannes Stephan. ✉email: roland.fischer@tum.de

The understanding of the physical and chemical properties of ligated metal clusters depends on the elucidation of their structures, which, however, relies on directed bottom-up wet chemical synthesis and suitable separation techniques to obtain defined, pure clusters[1–4]. Typically, ligated metal clusters cannot be accessed in a plannable retrosynthetic fashion[4–6]. Transient species in cluster growth reactions are usually highly reactive and therefore can follow multiple parallel pathways. This situation further complicates when more than one metal element and the associated chemical precursors are involved[7–13]. The mixed-metal cluster-generating reaction solutions contain several if not many species of very similar composition and structure (including intermediates and isomers) and the outcome of synthesis and separation are very dependent on the conditions. Targeting individual (stable) clusters from these solutions may be extremely difficult if not practically impossible as there are often too many limitations for setting and optimizing the experimental parameters in a trial-and-error fashion[5,8,9,14].

A well-studied example is the Ni/Ga system. Solid state Ni/Ga bulk phases, colloidal nanoparticles as well as small molecular complexes have been investigated in the context of their catalytic properties[15–18]. Ligated atom-precise, mixed-metal Ni/Ga clusters, however, are unknown to date. Certainly, this is a consequence of the difficult synthetic access of defined clusters. Is it possible to generate valuable information on individual species from complex cluster mixtures and avoiding separation? The herein-reported Ni/Ga cluster ensemble is quite characteristic and instructive of how to deal with and harvest from the richness of organometallic (all hydrocarbon ligated) cluster synthesis. Our methodology (Fig. 1) synergistically combines experiment and theory and bypasses the intrinsic limitations of isolating pure clusters from an ensemble of very closely related clusters $[Ni_{6+x}Ga_{6+y}](Cp^*)_6 = [Ni_xGa_{6+y}](NiCp^*)_6$ (1-5; $x + y \leq 2$; $Cp^* = C_5Me_5$). The key element is the assignment of chemical structures to each species of the ensemble by employing a computational permutative approach considering all possible isomers. As input for these calculations serve the elemental compositions, which are unequivocally identified by high-resolution mass spectrometry, including mass-tag (e.g. isotope) labeling as well as collision-induced fragmentation experiments. The result of this individual structural assignment is cross-validated by spectroscopic data and as well by reactivity studies of the whole ensemble.

## Results and discussion

**Synthesis.** The clusters are generated from the reaction of Ni(0) olefin complexes with $GaCp^*$ in toluene or mesitylene at elevated temperatures (70-110 °C; see SI, Supplementary Methods section). The reaction leads to the inseparable cluster ensemble $[Ni_{6/7/8}Ga_{6/7}](Cp^*)_6$, containing clusters of different metal atomicity (nuclearity): $M_{12}$, $M_{13}$ and $M_{14}$, respectively (Fig. 2). All products are very moisture sensitive and decompose immediately upon exposure to air. Crystalline material is obtained from saturated solutions at ambient temperature. Classical analytical techniques, however, completely failed in characterizing this material: Despite numerous SC-XRD measurements using different batches and crystallization conditions, the results of the refinement remain inconclusive (SI, Figure S24). Nevertheless, an octahedral $(MCp^*)_6$ shell around an inner metal core is conclusively obtained in all structural refinements. However, the exact Ni/Ga distribution within core and shell could not be deduced from the data sets. The elemental composition of solid bulk samples of the ensemble obtained by crystallization have been assessed by elemental analysis (C, H) as well as atomic absorption spectrometry (Ni, Ga) (SI, Table S24). These results were substantiated by XPS analysis (SI, Figure S27). In all samples, the Ni/Ga ratio is close to 1:1 with a slight bias to Ni. However, due the fact that all samples represent practically inseparable mixtures of several clusters, which are isolated as co-crystallites, the interpretation and assignment of the analytical data is not reasonable without independent quantification of the molar ratio of the clusters in the mixtures. Unfortunately, specific spectroscopic features are absent that would allow valid quantification of **1-5** in the ensemble.

**Mass spectrometric characterization.** Mass spectrometry (MS) is a powerful tool for analyzing clusters, especially electrospray ionization (ESI) MS has become a standard tool for the analysis of gold clusters in solution[19–21]. Whereas ESI is predominantly used for charged species, liquid injection field desorption ionization (LIFDI) is a particularly soft ionization technique for neutral compounds. For the title cluster ensembles, the identification of the reaction products is based on the accurate analysis of (in situ) mass spectra, which reveals the elemental compositions of the individual clusters. Sum formulas are accessible by high-resolution $m/z$ data of their molecular ions detected by LIFDI mass spectra as well as by careful analysis of their isotopic patterns (SI, Figures S11-13 &

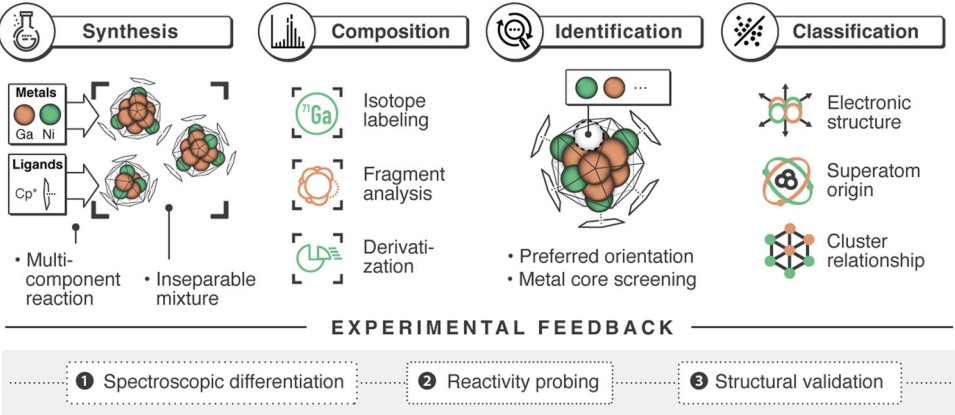

**Fig. 1 Schematic outline of the developed methodology.** After synthesis the compositions of each cluster in the ensemble are unequivocally assigned by high-resolution mass spectrometry (mass-tag labels and fragmentation experiments) and their respective structures are assessed by a computational permutative screening approach, taking all possible structures into account. The bonding situation of the species can then be explained through DFT calculations and the individual structure is further validated by an experimental feedback loop (spectroscopic characterizations and chemical reactivity towards probe substrates).

Tables S2-21). From the high-resolution mass spectra (Fig. 3), it can be deduced, that all clusters present in the solution contain exactly six Cp* ligands associated with different numbers of metal atoms: $Ni_6Ga_6$ (1), $Ni_7Ga_6$ (2), $Ni_6Ga_7$ (3), $Ni_7Ga_7$ (4) and $Ni_8Ga_6$ (5). The unambiguous assignment of sum formulas is achieved by labeling the clusters with $^{71}GaCp*$ and $GaCp*^{Et}$ ($C_5Me_4Et$), respectively, and relating the observed mass shifts to the number of Ga atoms or Cp* groups in the clusters (SI, Figure S16-18). We want to emphasize that all sample manipulations for recording spectra were performed under rigorous inert atmosphere to disclose any oxidation and/or hydrolysis[22]. Notably, all patterns are very broad and some overlap. The observed $m/z$ values point to the fact that during ionization the clusters loose hydrogen atoms: $[Ni_6Ga_6Cp*_6-2H]^+$ (1), $[Ni_7Ga_6Cp*_6-2H]^+$ (2), $[Ni_6Ga_7Cp*_6-H]^+$ (3), $[Ni_7Ga_7Cp*_6-$

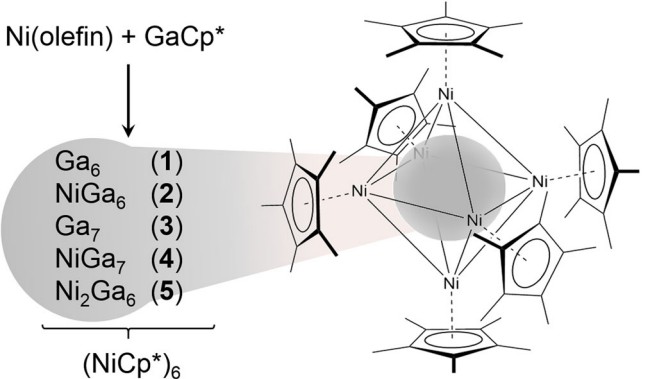

**Fig. 2 Synthesis of the cluster ensemble and its common structural motif.** Generation of the cluster ensemble, mainly consisting of $[Ni_6Ga_6](Cp*)_6$ (1), $[Ni_7Ga_6](Cp*)_6$ (2), $[Ni_6Ga_7](Cp*)_6$ (3), $[Ni_7Ga_7](Cp*)_6$ (4) and traces of $[Ni_8Ga_6](Cp*)_6$ (5). Compositions were established by LIFDI-MS. Structural assignment derived from a DFT screening approach of the full configuration space in combination with analytical and spectroscopic techniques.

$3H]^+$ (4) and $[Ni_8Ga_6Cp*_6-4H]^+$ (5). Such phenomena have been observed in field desorption mass spectra of organic[23,24] as well as cluster compounds[14,25] before. Notably, there is neither spectroscopic evidence for C-H activation or dehydrogenation in solution nor for the existence of H-atoms bound to the cluster core (*vide infra*). Collision experiments in a higher-energy collisional dissociation (HCD) cell allowed the distinction of the molecular ions from their fragment ions formed upon ionization during the mass spectrometric experiment (SI, Figure S19-23 & Tables S22-23). Species containing less than six Cp* ligands were identified as fragment ions, with fragmentation patterns showing the cleavage of Cp* as well as the splitting of $NiCp*_2$. The latter species is also detected as a molecular ion in the mass spectra. It should also be noted that clusters with slightly different $M_{12}$ and $M_{13}$ cores, in particular the species $[Ni_7Ga_5](Cp*)_6$ and $[Ni_8Ga_5](Cp*)_6$, were observed in traces, yet their small quantities did not allow for unambiguous assignment. By varying the Ni(0) olefin precursor (olefin: cod = 1,5-cyclooctadiene, dvds = 1,1,3,3-tetramethyl-1,3-divinyl-disiloxane), the qualitative composition of the cluster ensemble always remains the same, while the quantitative composition varies – $Ni(cod)_2$ leads to mixtures with $Ni_6Ga_7$ (3), while $Ni_2(dvds)_3$ gives mixtures with $Ni_7Ga_6$ (2) as the major component, respectively (SI, Figure S13). Notably, no significant differences between mass spectra from crystalline material or from reaction solution are observed (SI, Figure S14-15).

We want to emphasize that determination of the absolute quantities of the clusters is not feasible based on mass spectrometric peak intensities only, since identical ionizability of different clusters cannot be presupposed and a precise calibration would need pure isolated clusters as references, which are, however, not accessible. Nevertheless, evaluation of the LIFDI-MS data, including the labelling experiments unambiguously delivers the elemental composition of each component present in the ensemble. As an important first result we conclude that the clusters present in solution are all-Cp* ligand-protected and share the formula $[Ni_{6+x}Ga_{6+y}](Cp*)_6$ ($x + y \le 2$).

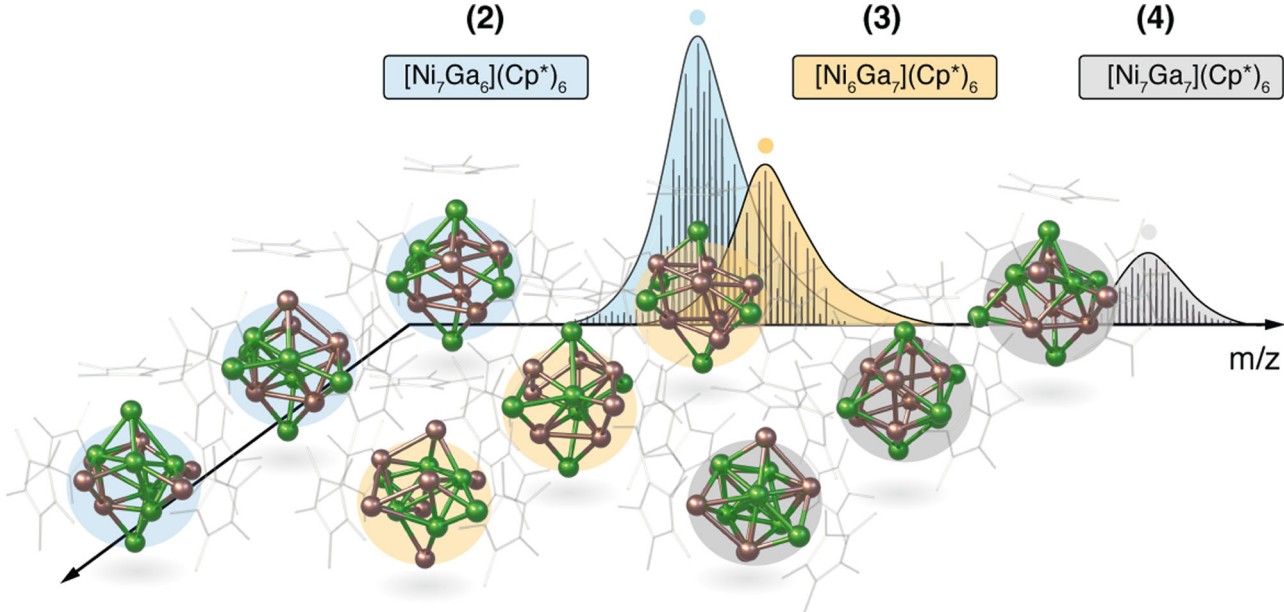

**Fig. 3 Visualization of the two levels of complexity (synthetic and analytical).** The LIFDI mass spectrum on the X-axis shows the major components of the cluster ensemble (2, 3 and 4; formulas and structures highlighted in blue, orange and grey, respectively). The organometallic, cluster-generating reactions never give a singular, pure compound (synthetic complexity). The individual elemental composition of each species can be unequivocally assigned by HRMS. The Y-axis shows DFT-based minima structures (only a few) of the isomers associated to each cluster composition (analytical complexity).

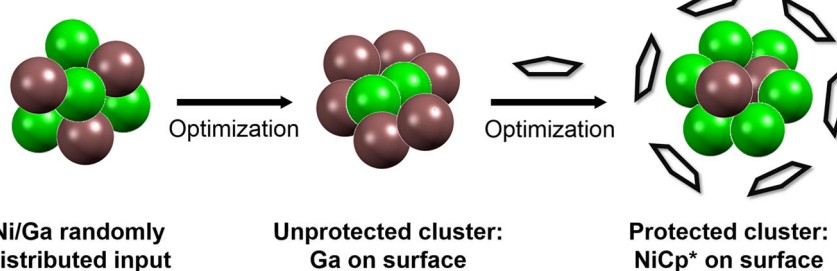

**Fig. 4 Schematic representation of the DFT screening approach.** Optimization of a large set of randomly distributed unprotected bimetallic (Ni: green, Ga: bronze) clusters followed by Cp* (grey) addition to the unprotected clusters, identifies Ni atoms to be located on the surface, attached to the Cp* ligands, in the clusters of the ensemble.

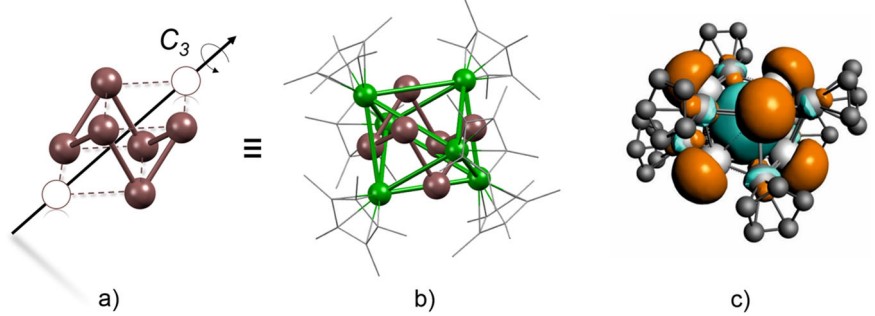

**Fig. 5 Bonding analysis of [Ga$_6$](NiCp*)$_6$. a** The distorted cubic parentage of the Ga$_6$ framework. **b** The DFT-optimized geometry of **1** showing a (NiCp*)$_6$ octahedron surrounding the Ga$_6$ framework. **c** electron density mapping of the LUMO of **1$^{Cp}$**, resembling the 2S orbital of a superatom.

**Structure assignment by computational screening.** For the assignment of chemical structures to the compositions of the clusters determined by mass spectrometry, a computational permutative approach has been developed using a methodology for total energy calculations (see SI, Supplementary Methods section). Important structural parameters derived from SC-XRD (SI, Figure S24), e.g. the octahedral arrangement of (NiCp*)$_6$ moieties, establish a framework for potential input structures for the calculations. Due to the large number of possible structural isomers, we established a systematic method for screening a large structural space of M$_{13}$ (**2** and **3**) and M$_{14}$ (**4**) clusters (SI, Figure S30). Our design principle essentially relies on the combination of two independent strategies: First, a set of monometallic clusters is obtained from gas-phase ab initio FHI-aims calculations (Fig. 4). The metal positions in these monometallic clusters are then randomly substituted by the second metal, resulting in a large set of bimetallic core structures. This scheme was applied with and without the presence of Cp* ligands placed on designed sites above the Ni/Ga clusters. The second strategy uses the crystallographically estimated core metal atom positions as input. By exchanging atom positions between four distinct chemical environments of the M$_{13}$ and M$_{14}$ cluster (SI, Figure S36) while keeping the ligand positions unchanged, a second large set of cluster geometries is created. The cluster structures obtained by these two strategies were combined and evaluated.

As a first step of the evaluation process, the whole set of optimized structures was screened for structural similarities with the target of reducing the number of clusters for further analysis. For this purpose a Hungarian algorithm, as well as an in-house modified Euclidian similarity distance algorithm were employed[26,27]. With the resulting set of unique cluster structures, we performed spin-polarized geometrical optimizations using FHI-aims with the PBE functional on the light-tier-2 basis set on each obtained structure and compared all structures in terms of relative energy (SI, Figures S29-S32). The most stable isomers of both the M$_{13}$ as well as M$_{14}$ naked clusters (i.e. without the Cp* protecting shell) show the Ni atoms on the structure's inner sites (Fig. 4; Figures S37, S39 and S41), following the radius order between the Ni and Ga atoms. The values for excess energy for nanoalloy formation is negative in all cases, i.e., their formation from the elements is thermodynamically favorable. Structural parameters as effective coordination number (ECN), average distances (d$_{av}$), magnetic moments and chemical order ($\sigma$) were obtained (SI, Figures S38, S40 and S42); however, there is not an apparent trend in these indicators if compared to the relative energy of the isomers. As the Cp* ligands are added to the calculations, we observe key changes on the favored Ni and Ga positions. Now, the most stable isomers show the Ni atoms on the outermost sites of the clusters (Fig. 5b). The Cp* ligands coordinate to these Ni sites (Fig. 4; Figures S43, S45-47, S50). Notably, their overall topology is fully consistent with the SC-XRD data (*vide infra*). Especially the M-Cp* distances in the experimental structure point to M being rather Ni than Ga[28–32]. Once the metal atoms at the Cp* binding are exchanged to Ga atoms, an energy increase up to 5 eV (M$_{14}$) or 4 eV (M$_{13}$) is observed. The magnetic moments are 1 for the metal-cores Ni$_7$Ga$_7$ and Ni$_6$Ga$_7$ but are 0 for Ni$_7$Ga$_6$. Notably, this is well consistent with the diamagnetic nature of Ni$_7$Ga$_6$ concluded from SQUID measurements (*vide infra*). Charge analysis shows a slight charge transfer from the metal core to the Cp* ligands with a correlation of the relative energies and average distance from Cp* to the metal core. In summary: Our computational ab-initio approach, investigating all possible compositional isomers [Ni$_{6/7}$Ga$_{6/7}$](Cp*)$_6$, reveals that the most stable cluster isomers are all built upon the same structural prototype: a pure Ga$_6$ or NiGa$_{6/7}$ inner core, surrounded by a monometallic (NiCp*)$_6$ shell. This is well consistent with the interpretation of SC-XRD data, but we can further support this assumption by spectroscopic data, in particular $^{13}$C MAS NMR, vibrational spectroscopy and SQUID (*vide infra*).

The cluster-generating reaction obviously includes complete Cp* transfer from Ga to Ni. This is a common transmetalation pattern in the reaction of ECp* (E = Al, Ga, Zn) involving open d-shell transition metals and has also been observed in reactions of GaCp* with organometallic complexes of nickel[28,33], cobalt[34], iron[34,35], and rhodium[36,37]. The driving force of this Cp* transfer step is related to the formation of thermodynamically stable, closed-shell half-sandwich transition metal complexes with strong TM-Cp* interactions (favored by an open d-shell TM interacting with Cp*). As discussed in the computational section, this reasoning also applies to this case.

**Structure validation by spectroscopy**. In contrast to mass spectrometric analysis, spectroscopic information does not address individual species directly but concerns the whole cluster ensembles. Dependent on the synthetic conditions, ensembles enriched with either **2** or **3** can be produced (vide supra). SQUID/EPR, as well as Raman (SI, Table S1) provide data to support the compositional and structural assignment of the cluster metal cores: SQUID of samples enriched with the intrinsically paramagnetic $Ni_6Ga_7$ (**3**) indeed show stronger paramagnetism, whereas the corrected SQUID data of samples enriched with diamagnetic $Ni_7Ga_6$ (**2**) exhibit a "cusp" at 220 K, which is indicative for samples with a large diamagnetic component (SI, Figure S28). EPR measurements confirm this interpretation: In all samples a signal centered at g = 2.13 is observed, which is considerably more intense in samples enriched in the paramagnetic cluster **3** (SI, Figure S29). Raman spectra point to significantly different metal core connectivity in **2** and **3** (the skeletal Ni-Ga and Ga-Ga stretching modes are below 250 cm$^{-1}$). FT-IR and Raman spectroscopic data of the cluster mixtures (ATR, solid state; SI, Figures S6-10) confirm the very symmetric and uniform shielding of the metal core by intact Cp* units (Fig. 2). No evidence is present for any intramolecular bond activation of the Cp* ligands. A detailed assignment of all IR and Raman signals is given in the SI (Table S1) and has also been recently reported and discussed in detail[38].

The most valuable information on the ligand shell of the clusters is derived from $^{13}C$ NMR spectra of cluster ensembles: The (Cp*)$_6$ ligand shell gives rise to a single set of signals at 98.10 and 12.77 ppm which correspond to the ring carbon atoms and the methyl groups of Cp*, respectively (SI, Figure S2). This strongly indicates the presence of only one metal type coordinated to the Cp*, therefore a "homometallic" (MCp*)$_6$ shell. The observed chemical shift at 98.1 ppm is typical for NiCp* groups measured in solution[30,39] while GaCp* signals are typically found at 110-115 ppm[29,31,32].

In addition, we confirmed the assignment of GaCp* vs. NiCp* by comparison with the $^{13}C$ MAS NMR spectrum of the [$Ni_4Ga_3$](Cp*)$_3$(dvds)$_2$ cluster[28] (SI, Figure S3), which contains both moieties. In summary, all spectroscopic data are well consistent with the computationally derived structures, thus validating the structure assignment strategy via a DFT screening approach.

**Bonding analysis**. To further investigate and rationalize the cluster structures from a bonding point of view and to analyze common features as well as subtle differences in their electronic situations, we optimized the structures of the model series [$Ni_{6/7/8}Ga_{6/7}$](Cp)$_6$ by means of DFT calculations at the BP86/TZ2P level of theory (see Computational Details, SI). The use of simple $C_5H_5$ (Cp) in place of the real $C_5Me_5$ (Cp*) ones allowed a straightforward analysis of their electronic structures. In a second step, the [$Ni_{6/7/8}Ga_{6/7}$](Cp*)$_6$ series was also optimized and compared to its Cp counterpart. All the structures discussed below are true energy minima confirmed by frequency

calculations and all even-electron species show a significant, although not very large, HOMO-LUMO gap. Selected computed data are given in the SI, Table S25. Over the course of our calculations, we identified [$Ni_6Ga_6$](Cp)$_6$ (**1$^{Cp}$**) as a base frame, as it is part of all other structures. Although **1** appears as a minor component of the cluster ensembles, at least based on LIFDI-MS data, **1$^{Cp}$** will be our "reference" species in the following bonding analysis.

The lowest energy structure of **1$^{Cp}$** can be described as an octahedral (NiCp)$_6$ outer shell encapsulating an inner $Ga_6$ "cube" having two missing vertices that are situated along one of its solid diagonals (Fig. 5a). In such a "cubic" configuration, every square face of the cube is made of three Ga and one vacant vertex. Each of the six faces of the cube are capped by one (NiCp) unit (Fig. 5b). Interestingly, such cubic core arrangements were often found as structural motifs in the screening approach (vide supra). The bonding within this $Ga_6Ni_6$ core is ensured by 36 electrons, three provided by each Ga atom and three by each $Ni(\eta^5-C_5R_5)$ fragment. Neglecting the long Ga···Ga contacts (~ 3.0 Å), the 36 electrons can be assigned to the 18 Ni-Ga bonds. Within this localized 2-center/2-electron bonding description, the Ni centers obey the 18-electron rule, while the Ga atoms feature a sextet electronic environment, which is a reasonably stable situation for group 13 elements, although somehow electron-deficient. This electron deficiency is associated with the presence of a rather low-lying vacant sp hybrid orbital on each individual Ga center. The six vacant Ga sp hybrids combine and their lowest (thus bonding) combinations mix somewhat with occupied Ni-Ga orbitals, thus conferring some Ga···Ga through-bond attractive interaction, as exemplified by the Ga···Ga "non-bonding" contacts (SI, Table S25). The LUMO of **1$^{Cp}$** is the in-phase combination of the six Ga sp hybrids (Fig. 5c). With substantial Ga···Ga overlap inside the $Ga_6$ cage, it resembles the 2S orbital of a superatom[40]. The computed HOMO-LUMO gap of **1$^{Cp}$** (0.65 eV) is indicative of reasonable chemical stability. Thus, [$Ni_6Ga_6$](Cp)$_6$ can be described as an intermetallic superatom complex.

The electronic structure of the superatom base cluster [$Ni_6Ga_6$](Cp)$_6$ (**1$^{Cp}$**) allows for the integration of one or two additional metal atoms resulting in the extended derivative clusters [$Ni_7Ga_6$](Cp)$_6$ (**2$^{Cp}$**), [$Ni_6Ga_7$](Cp)$_6$ (**3$^{Cp}$**), [$Ni_7Ga_7$](Cp)$_6$ (**4$^{Cp}$**) and [$Ni_8Ga_6$](Cp)$_6$ (**5$^{Cp}$**). The optimized structure of [$Ni_7Ga_6$](Cp)$_6$ (**2$^{Cp}$**) (SI, Figure S52a) can be described as resulting from the occupation of one of the "missing" cube vertices in **1$^{Cp}$** by one additional ("exposed") Ni atom (SI, Table S25 and Figure S52c). Its bonding to the **1$^{Cp}$** fragment results from two components. One is associated with a 3d($Ni_{exp}$) donation into the accepting orbitals of **1$^{Cp}$** discussed above (1.05 electron). As a result, the superatomic 2S-type LUMO of **1$^{Cp}$** is now completed by a $d_{z2}(Ni_{exp})$ contribution in **2$^{Cp}$** (SI, Figure S52b). The other bonding component results from a similar (1.05 electron), but backward, electron transfer into the 4s/4p AOs of the "exposed" Ni from its three Ni neighbors.

Adding a Ga atom to the base cluster **1$^{Cp}$** results in [$Ni_6Ga_7$](Cp)$_6$ (**3$^{Cp}$**). **3$^{Cp}$** is an odd-electron species, for the sake of simplicity we investigate first its closed-shell cation [**3$^{Cp}$**]$^+$. It is obvious that the single Ga atom in **3$^{Cp}$** is substantially weaker bonded to **1$^{Cp}$** than the single Ni atom in **2$^{Cp}$**, as exemplified by long $Ga_{exp}$-Ga (3.131 Å) and $Ga_{exp}$-Ni (3.298 Å) distances (SI, Figure S52c and Table S25). This is consistent with the fact that now there is negligible electron transfer from Ga to the **1$^{Cp}$** fragment. The major electron transfer occurs between the 3d(Ni) and the vacant 4p orbitals of the "exposed" Ga$^+$ (0.83 electron). Therefore, this "exposed" atom does not participate to the 2S-type LUMO of [$Ni_6Ga_7$]($C_5R_5$)$_6$$^+$ (R = H, Me), which resembles that of the uncapped **1$^{Cp}$**. Going now to the "real" neutral,

odd-electron, $3^{Cp}$, barely changes the cluster structure, which nicely underlines the electronic flexibility of the base cluster $1^{Cp}$. Occupying the 2S-type LUMO of the cation cluster by a single electron results only in some shortening of the Ga-Ga distances within the $1^{Cp}$ cage, in line with the Ga···Ga bonding character of this orbital (SI, Table S25). Calculations show that it is possible to cap the $1^{Cp}$ cage on both sides of the $C_3$ axis by two atoms of Ni and/or Ga nature, thus completing the cube of Fig. 5a. The data obtained for $4^{Cp}$ and $5^{Cp}$ (SI, Table S25) show similar characteristics as their monocapped relatives.

The computed Cp* series of clusters 1-3 provided quite similar results as that of their Cp homologues (SI, Table S25). They exhibit slightly lower HOMO-LUMO gaps. This is in line with their slightly shorter Ga···Ga contacts. This is also consistent with the fact that their potential energy surfaces were found to be quite flat and their equilibrium structure less symmetrical that of their Cp counterparts. In particular, concerning 3 the "exposed" Ga atom is connected to one, rather than three Ni atoms as in $3^{Cp}$.

It is of note that the presence in solution of alternative low-energy isomers of 1-3 is not to be fully excluded. As already mentioned in the previous section, they all show the same feature, namely a Ga or Ga/Ni core embedded in an outer $(NiCp*)_6$ shell. This common structural characteristic makes all of them having the same superatomic electronic structure as those of the 1-3 most stable isomers analyzed above. It follows that interconversion between isomers in solution is not also to be fully excluded.

To conclude, the electronic structures of clusters 1-3 are strongly related to each other. Thus, clusters 2 and 3 are formed by addition of Ga or Ni atoms to the inner $Ga_6$ core of the superatomic base cluster 1, without remarkable structural deviations of the metal core. In some way, this may metaphorically be linked to the fact that these clusters occur experimentally only in ensembles, preventing targeted synthesis of single cluster species by the organometallic route.

**Structure-Reactivity Relationship**. Despite their strong electronic and structural similarities, the clusters should be markedly different in their chemical reactivity. Thus, clusters 2, 4 and 5

contain naked Ni atoms in the inner core, whereas clusters 1 and 3 contain Ga atoms only. In order to investigate whether this structural difference is reflected in the reactivity of the singular species of the ensemble, we exposed the cluster mixtures to CO as a probe molecule for naked Ni atoms. Pressurizing respective samples in toluene-$d_8$ with 2.5 atm CO at ambient temperatures results in dark red solutions after 120 min. Careful analysis of the LIFDI mass spectra point to a significant reactivity difference of the four major cluster species 1-5, (SI, Figures S58-61): While the patterns of $[Ni_6Ga_6](Cp*)_6$ (1) and $[Ni_6Ga_7](Cp*)_6$ (2) are virtually unaffected by the presence of CO, the intensity of pattern assignable to the $M_{13}$ cluster $[Ni_7Ga_6](Cp*)_6$ (2) decreases considerably, indicating a high affinity of this species towards CO (SI, Figures S60-61). The pattern of the $M_{14}$ cluster $[Ni_7Ga_7](Cp*)_6$ (4) is still detectable in the presence of CO, however, in addition its CO adduct $[(CO)NiNi_6Ga_7](Cp*)_6$ is observed at $m/z = 1734.4$. This striking reactivity difference of the four species nicely correlates with the presence of active nickel atoms in the cluster core (Fig. 6). A more detailed analysis of all identified species and cluster degradation products induced by CO can be found in the SI (Figures S53-61). By DFT calculations stable adducts for the interaction of carbon monoxide with $[Ni_7Ga_6](Cp*)_6$ (2) and $[Ni_7Ga_7](Cp*)_6$ (4) were found. The CO binds to the exposed Ni atom of the inner core, with rather short Ni-CO distances of 1.774 and 1.770 Å, respectively. The CO lone pair interacts inter alia with the superatomic 2S-type LUMO, which has $Ni_{exp}$ character. The cluster structures are little affected by the CO coordination.

As a second probe molecule for screening cluster reactivity, (triisopropylsilyl)acetylene (TIPSA) was used. Alkynyl ligands are widely employed as stabilizing ligands for metal clusters[41–43]. In fact, the reaction turned out to be highly selective, allowing separation of a singular species: The cluster ensemble is treated with an excess of TIPSA under irradiation (350 nm) and the cluster $[Ni_8Ga_6](Cp*)_6(TIPSA)_2$ was isolated in pure single crystalline form. Notably, only $[Ni_8Ga_6](Cp*)_6$ (5) reacts under these conditions, whereas all other clusters are unaffected by this reagent. Here, an unambiguous assignment of the structure, including all metal atom positions, is possible (Fig. 7). Indeed, in agreement with all results presented above, the Cp* ligands are

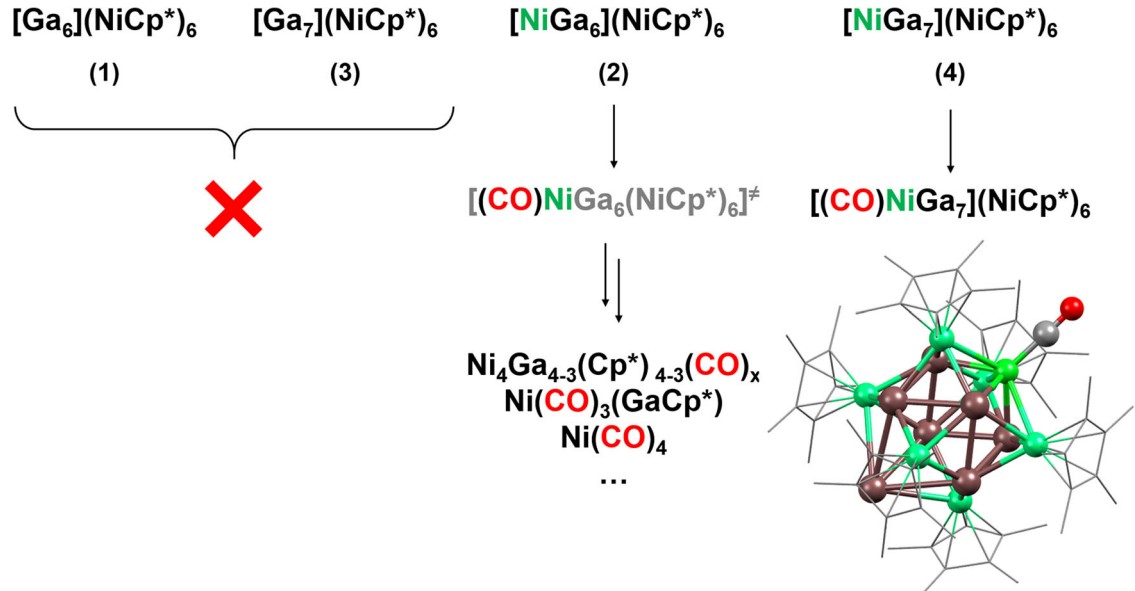

**Fig. 6 Structure-activity relationship of Ni/Ga clusters towards the coordination of carbon monoxide.** Whereas clusters with accessible nickel atoms in their core form adducts with CO, clusters with exclusively gallium cores do not react. For $[NiGa_7](NiCp*)_6$ (4), the adduct with CO (see calculated structure on the bottom right) could be detected by LIFDI MS whereas in the case of $[NiGa_6](NiCp*)_6$ (2) only smaller (cluster) fragments are found. For more details see SI, Figures S51-S59.

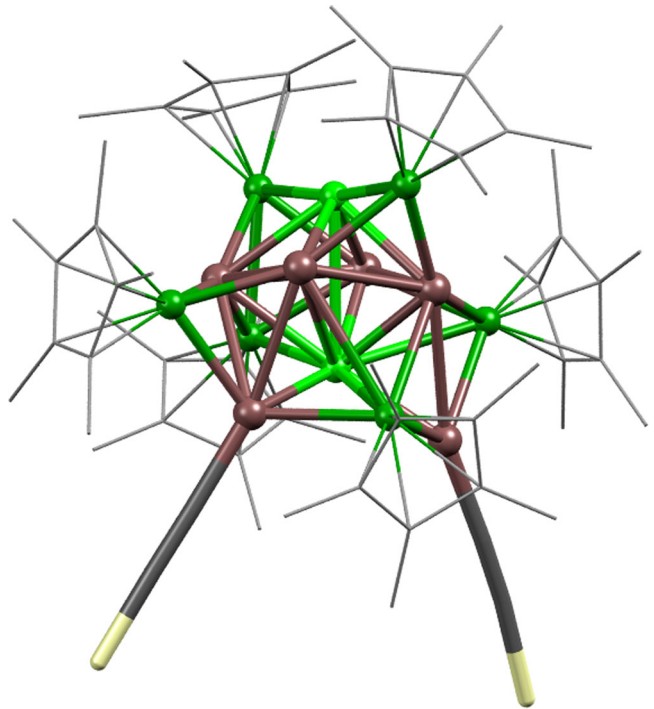

**Fig. 7 Molecular structure of [Ni$_8$Ga$_6$](Cp*)$_6$(TIPSA)$_2$.** All atom positions are unambiguously assigned. All Cp* ligands are attached to Ni, TIPSA ligands (rod-like presentation) are bound to Ga. Ni attached to Cp* (dark green), Ni (light green), Ga (bronze), Si (yellow). Cp* in wireframe, isopropyl groups, H atoms and solvent molecules are omitted for clarity.

attached to Ni atoms only, while the two alkynyl ligands are attached to Ga atoms and not to Ni (as the CO would). A full spectroscopic analysis and discussion of the structural details of this derivative, together with insights into its formation mechanism are subjects of ongoing research.

**Conclusions and Perspective.** Our case study on the superatomic cluster ensembles [Ni$_{6+x}$Ga$_{6+y}$](Cp*)$_6$ (x + y ≤ 2) demonstrates how to gain valuable structure-property relationship data without isolating analytically pure closely related clusters from practically inseparable reaction solutions. Our methodology uses mass spectrometry for determination of cluster compositions from raw synthetic mixtures and relies on advanced computational methods for the elucidation of chemical structures. Organometallic cluster synthesis creates a chemical complexity of parallelly formed intermetalloid clusters, each with an individual composition, structure and reactivity. A second level of complexity is related to the analysis of each individual in an (almost) unlimited space of possible isomers (shape, connectivity, element distribution). An advanced computer assisted approach allows us to find a solution for the latter (analytic complexity), which in turn enables us to fully embrace the first (synthetic complexity). In other words, the exact chemical structures of a variety of coexisting products are accessible by a synergistic combination of experiment and theory. Based on the presented methodology structures can be unambiguously assigned. Structural rationalization then gives insight into the bonding situation and enables the design of experiments to selectively obtain and eventually isolate derivatives of some of the original clusters in pure form. The introduced "ensemble approach" suggests an avenue for exploiting the rich chemical space of organometallic chemistry for intermetalloid clusters. We anticipate the discovery of interesting, potentially novel reactivities of so far unknown, and larger intermetalloid clusters and reactive superatom complexes.

## Methods

**General.** Detailed experimental procedures, analytical, spectroscopic and computational techniques and data are given in the Supporting Information (see Supplementary Methods section), including solution NMR & MAS NMR spectroscopy (Figures S1-S5), vibrational spectroscopy (Figures S6-S10), LIFDI mass spectra (Figures S11-18), ion fragmentation experiments (Figures S19-S23), SC-XRD (Figure S24), PXRD (Figure S25-26), XPS, EPR & SQUID (Figures S27-S29), DFT calculations (Figures S30-S52) and reactivity tests with carbon monoxide (Figures S53-S59). Elemental analyses and atom absorption spectrometry were performed by Mikroanalytisches Laboratorium Kolbe, Oberhausen, Germany. Mass spectrometric data of diluted toluene solutions of the cluster ensemble were acquired using a setup comprising a ThermoFisher Exactive Plus Orbitrap mass spectrometer equipped with a liquid injection field desorption ionization (LIFDI) source by Linden CMS GmbH, connected to a glovebox[22].

**Synthesis of the cluster ensemble with major component (3).** [Ni(cod)$_2$] (500 mg, 1.18 mmol) was suspended in toluene (5 ml) and GaCp* (435 mg, 2.12 mmol) was added at room temperature. The dark solution was heated to 65 °C for two days. Upon slow cooling to room temperature, a black solid was obtained, which was extracted with hot (100 °C) mesitylene (16 ml). The solution was concentrated under reduced pressure, isolated by means of cannula filtration and dried under reduced pressure to yield the cluster ensemble as black crystalline solid (155 mg).

**Synthesis of the cluster ensemble with major component (2).** GaCp* (482 mg, 1.07 mmol) was added to a solution of [Ni(GaCp*)(dvds)] (290 mg, 1.42 mmol) in toluene (2 mL). After heating the dark red reaction solution to 110 °C for 3 h, the hot solution was filtered via a cannula. The solution was allowed to cool to ambient temperature overnight and a black precipitate was formed. This was separated from the reaction solution by means of cannula filtration. The residue was washed with small amounts of cold *n*-hexane and dried under reduced pressure, to yield a black crystalline solid (60.8 mg). The synthesis can be performed also with [Ni$_2$(dvds)$_3$] instead of [Ni(GaCp*)(dvds)] while the Ni/Ga ratio is kept constant. Work up, yield, and spectroscopic results are identical.

**Density Functional Theory calculations.** Total energy Calculations were based on spin-polarized density functional theory (DFT) within the semilocal exchange-correlation energy functional proposed by Perdew–Burke–Ernzerhof (PBE). The Kohn–Sham (KS) orbitals were described by numeric atom-centered orbitals (NAO), as implemented in the all-electron Fritz–Haber Institute ab initio molecular simulations (FHI-aims) package. We employed a minimal NAO basis set with a set of additional NAO added hierarchically up to the second basis set improvement, called light-tier2 in FHI-aims notation. For relativistic corrections, the scalar-relativistic framework with zero-order regular approximation (ZORA) was employed. For the self-consistency solution of the KS equations, we employed a total energy criterion of $10^{-5}$ eV, while the equilibrium geometries were obtained once the atomic forces were smaller than $10^{-2}$ eV Å$^{-1}$. For the vibrational frequency calculations, we decreased the forces criteria for $10^{-4}$ eV Å$^{-1}$ to calculate the Hessian matrix elements using atomic displacements of $2.5 \times 10^{-3}$ Å. To avoid fractional occupation of the highest occupied molecular orbitals (HOMO) and lowest unoccupied molecular orbitals (LUMO), we employed a Gaussian broadening of 1 meV. For bonding analysis, Density Functional Theory (DFT) calculations were carried out with the use of the Amsterdam

Density Functional code (ADF2017) with the addition of Grimme's D3 empirical corrections. The triple-ξ Slater basis set plus two polarization functions (STO-TZP) was used, together with the Becke-Perdew (BP86) exchange-correlation functional. All the optimized structures were confirmed as true minima on their potential energy surface by analytical vibration frequency calculations.

## Data availability

The authors declare that the data supporting the findings of this study are available within the paper and its supplementary information files or from the corresponding authors on reasonable request.

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

## Acknowledgements

We thank Dardan Ukaj for the Raman measurements and Dr. Alexander Pöthig and Dr. Wilhelm Klein for their help in the interpretation of the SC-XRD data. We would like to express our thanks to Fabian Schmidt and Dr. Gabriele Raudaschl-Sieber for recording the solution and solid-state NMR spectra, respectively. Finally, we want to acknowledge the contribution of Tobias Steinke, Jana Weßing and Julius Hornung who have laid the scientific foundations as part of their dissertation theses on which the discoveries in this work are based. This work was funded by the German Research Foundation (DFG) within a Reinhard Koselleck Project (FI-502/44-1) and the TUM Graduate School. TUM Global Incentive Fund is acknowledged for a travel fund for M.M.; S.K. and J.-Y.S. are grateful to GENCI (Grand Equipment National de Calcul Intensif) for HPC resources (Project A0050807367). The authors gratefully acknowledge support from FAPESP (São Paulo Research Foundation, Grant Numbers 2017/11631-2 and 2018/21401-7), Shell and the strategic importance of the support given by ANP (Brazil's National Oil, Natural Gas and Biofuels Agency) through the R\&D levy regulation.

## Author contributions

R.A.F., M.C. and C.G. supervised the project. M.M., J.S., L.S., K.H., M.S. and P.H. carried out the experimental work. C.J. collected SC-XRD data, T.K. and S.G. performed XPS analyses. D.H. assisted in mass spectrometric data analysis. S.K. and J.Y.S. as well as A.C.H. D.S. and J.L.F.D.S. carried out DFT calculations and bonding analyses. O.C. performed SQUID and EPR and J.M. vibrational spectroscopy experiments.

## Funding

## Competing interests

The authors declare no competing interests.
