## [Peer Review File · Communications Chemistry]

Reviewers' comments:

Reviewer #1 (Remarks to the Author):

Prof. Fischer and group have reported an interesting finding based on NiGa clusters with Cp ligands. They looked beyond the conventional synthetic pathways and studied in detail with what was supposed to be discarded. They have used LIFDI mass spectrometry in combination with DFT calculations to shed light into the structures of the clusters with unsolved crystal structures. They have also used the fragment ions to conclude their claim. Overall the manuscript is well-written and the data are adequately described. I recommend publication of the article with minor revision noted.

My specific comments are as follows:

1. Although the authors claimed the position of the Ni and Ga atoms from DFT and compared the data acquired from MS, it is not always true that the gas phase structures will be similar as their condensed phase structure. Please elaborate this concern in the revised manuscript.
2. From DFT, the isomeric clusters are very close in energy with several isomers within 1 eV range which is not so high for 13 or 14 atom systems. How the authors are discarding the other isomers based on the presented experimental data is not yet clear to me.
3. There could be completely different structure with 13 and 14 atom cluster core. The SC-XRD data may justify the structures reported for (NiGa)₁₄(Cp)₆ but for (NiGa)₁₃ the same correlation might not hold correct.
4. Is it possible to experimentally figure out the presence of other isomeric forms of the clusters reported here? Have the authors considered measuring ion mobility spectra of the ions? Comparing the experimental and calculated collision cross sections of the ions with the claimed structures may confirm the reported structures. Also, high resolution IMS can help in separating close lying isomeric forms.
5. Please explain the origin of fragment ion Ni(COD)₂ and the location of the COD ligands in the cluster structures.
6. Please show the MS/MS data and plot the fragment ion intensity alongside the parent ions in the fragmentation related plots. There is nothing mentioned on the voltages used for the CID experiments.

Also, mention the formula and m/z of the parent ions on each of the graphs.

7. Most of the ESI figures are randomly given and not properly mentioned in the main text.

8. Minor comment: There are several typos throughout the manuscript.

Reviewer #2 (Remarks to the Author):

Fischer et al. described systematic analyses of heterometallic clusters formed in the reaction mixture by several analytical techniques as well as DFT study to optimize the possible structure with monitoring their stabilities. As the authors emphasized in the introduction, isolation of each component during the synthesis of heterometallic clusters was difficult, and in most cases, researchers tried to isolate one component by recrystallization, meaning that no further investigation on the rest of the components. Recent investigation on the heterometallic clusters by ESI-MS analyses is indicative for identifying the molecular weight and composition, but no structural information was obtained. Combination of all the analytical techniques, DFT, and some reactive study is essentially important to find out which types of metal clusters are present in the reaction mixture. This reviewer is not a specialist for the systematic computational screening and analyses of the possible structures and compositions, but it is reasonable for finding out the relation between the molecular weight and optimized structure using DFT based on the structurally similar clusters. In addition, deep investigation on the ESI-MS analyses using isotope and modified supporting ligands is reasonable to confirm the number and composition of metal atoms. This approach is essentially very supportive for the researchers in the field of organometallic and cluster chemistry because they are always struggling for the purification of the target product; in this regard, this reviewer raises one essential point on the reactive study. In this current form, it is quite difficult to follow the fate of each cluster after exposing to CO gas and silylacetylene. The authors should summarize which cluster was consumed by the MS analyses, and include one Scheme or Figure in the main text to find out the relationship of the reactive clusters, byproduct, and unreactive clusters (as a minor revision request); otherwise, it is difficult to follow the reaction sequences.

Reviewer #3 (Remarks to the Author):

Roland A. Fischer and co-authors describe a rather comprehensive, multi-technology approach to investigate reaction mixtures containing polyatomic Cp*-decorated Ni/Ga clusters. The class of compound has been investigated by the Fischer group for many years.

As outlined in their introduction, the elucidation of species being present in mixed-metal cluster mixtures is both difficult and important for learning more about the cluster growth mechanisms and about the nature of the potentially reactive species in solution. Beyond this background, the study is a nice contribution to intermetalloid cluster research.

The manuscript is well written, and the results are pretty well documented, but in contrast to the statement given in the abstract, even a very thorough study of the reaction mixtures will not “overcome[s] the intrinsic limitations of poor selectivity of mixed-metal cluster syntheses and

separation". This statement is at least misleading, as the authors probably want to say that "one can overcome the lack of information" in such cases. More importantly though, this reviewer does not think that the study should be reported as a conceptually new approach for diverse reasons, and therefore requires major revision:

1) The combination of HRMS studies and DFT calculation to study reaction mixtures of mixed-metal clusters (with or without ligand shell), referred to as cross-validation here, is a well-established approach, see recent works by groups working in the field of mixed-metal Zintl clusters, e.g., the groups of Sun/McGrady and Dehnen/Weigend.

2) Following the description of the computational work, we consider the method not suitable to scan the entire structural space; the statement starting with "Our computational ab-initio approach, screening the complete structural space...reveals that the most stable cluster isomers are all built upon..." is thus not yet proven to apply to our understanding.

3) It also remains questionable whether the computational screening method is suited to generally detect the global minimum structure, especially for any system beyond the reported, very specific, Cp*/Ni/Ga clusters. The closing remark "We anticipate the discovery of [...] and so far unknown, and larger intermetalloid clusters and reactive superatom complexes" is thus far-fetched.

4) An alternative approach that does not rely on the knowledge of a crystal structure, makes use of the combination of a (very quick) calculation using first order perturbation theory in the nuclear charge and a genetic algorithm study. We assume that the methodology reported here is more complicated (thus, more time-consuming), not fully comprehensive, and not easily transferrable to other elemental combinations or unknown structures – unless demonstrated on further examples. The other method should be applied (at low costs) for comparison.

This reviewer thinks that the novelty and generalizability of the combination of methods, and the DFT study in particular, is not as high as implied. Based on the published literature (in part cited in this manuscript), we do not consider this combination of analytical methods conceptually new, although properly applied to the specific cluster system. A more efficient computational method should be applied for comparison before claiming the advantage of the twofold study described herein.

The authors should tone down all statements that include terms like "novel" or "complete", before the manuscript can be reconsidered for publication. Whether it still fulfills the requirements for publication in a high-ranking journal such as Communications Chemistry should be reevaluated thereupon.

Technische Universität München | LS f. Anorg. u. Metallorg. Chemie
Lichtenbergstr. 4 | 85748 Garching

Communications Chemistry

Editorial Office

- electronically submitted -

Garching, November 6th, 2023

Revision for article: "**Intermetalloid Cluster Ensembles: A Ni/Ga Case Study**", Maximilian Muhr, Johannes Stephan, Lena Staiger, Karina Hemmer, Max Schütz, Patricia Heiß, Christian Jandl, Mirza Cokoja, Tim Kratky, Sebastian Günther, Dominik Huber, Samia Kahlal, Jean-Yves Saillard,* Olivier Cador, Augusto C. H. Da Silva, Juarez L. F. Da Silva,* Janos Mink, Christian Gemel, and Roland A. Fischer*

Dear Editors and Referees,

Thank you for considering our manuscript for publication as a regular article in *Communications Chemistry*. We have considered all the reviewers' critics and suggestions in this revised version of the manuscript. Below, we address every comment and provide a summary of the changes made both in the manuscript and the supporting information. Furthermore, we added also the marked manuscript, where blue indicates new text and red removed text.

Reviewer 1:

"Although the authors claimed the position of the Ni and Ga atoms from DFT and compared the data acquired from MS, it is not always true that the gas phase structures will be similar as their condensed phase structure. Please elaborate this concern in the revised manuscript."

First of all, for a very large majority of "molecular" clusters where both an accurate X-ray structure and a gas-phase DFT-optimized geometry exist, the two structures are similar.^{1,2} Moreover, as detailed just below in the answer to the next question, all the computed low-energy isomers of the computed species exhibit the same (core)@(Ni₆Cp*₆) feature, which is also that coming out from the SC-XRD data (Figure S24). This common feature makes all the low-energy structures electronically similar and we have a good orbital rationalization for their stability (superatomic description, see below).

"From DFT, the isomeric clusters are very close in energy with several isomers within 1 eV range which is not so high for 13 or 14 atom systems. How the authors are discarding the other isomers based on the presented experimental data is not yet clear to me."

As already mentioned in the "Structure Assignment by Computational Screening" section, the most stable isomers are all showing the same feature, namely a Ga or Ga/Ni core embedded in an outer (NiCp*₆) shell. This common structural characteristic makes all of them having the same superatomic electronic structure as those of the **1-3** most stable isomers analyzed in the "Bonding Analysis" section. A sentence, which is also a contribution to the answer to the previous question (just above), has been added at the end of this section, accordingly.

"There could be completely different structure with 13 and 14 atom cluster core. The SC-XRD data may justify the structures reported for (NiGa)₁₄(Cp)₆ but for (NiGa)₁₃ the same correlation might not hold correct."

In principle, the concern of reviewer 2 seems intuitive, especially if the number of metal atoms within a cluster differs. However, SC-XRD data of co-crystallised Ni/Ga clusters (namely [NiGa₆](NiCp*₆) and [Ga₇](NiCp*₆) show a *compa-*

1 Crasto, D., Malola, S., Brososky, G., Dass, A. & Häkkinen, H. Single Crystal XRD Structure and Theoretical Analysis of the Chiral Au₃₀S(S-*t*-Bu)₁₈ Cluster. *J. Am. Chem. Soc.* 136, 5000-5005, doi:10.1021/ja412141j (2014).

2 Campos, J., Sharninghausen, L. S., Crabtree, R. H. & Balcells, D. A Carbene-Rich but Carbonyl-Poor [Ir₆(IMe)₃(CO)₂H₁₄]²⁺ Polyhydride Cluster as a Deactivation Product from Catalytic Glycerol Dehydrogenation. *Angew. Chem. Int. Ed.* 53, 12808-12811, doi:https://doi.org/10.1002/anie.201407997 (2014).

Technische Universität München
TUM School of Natural Sciences
Lehrstuhl für Anorganische und
Metallorganische Chemie (AMC)

Prof. Dr. Roland A. Fischer
Lichtenbergstraße 4
85748 Garching b. München

Sekretariat:
Rodica Dumitrescu
Martin Schellerer
Dr. Dana Weiß

roland.fischer@tum.de
www.ch.nat.tum.de/amc/home/
www.tum.de

sekretariat.amc@tum.de
Tel. +49 89 289 13081
Fax +49 89 289 13194

Bayerische Landesbank
IBAN-Nr.:
DE1070050000000024866
BIC: BYLADEMM
Steuer-Nr.: 143/241/80037
USt-IdNr.: DE811193231

rably uniform distribution of NiCp* moieties in a quasi-octahedral array around the cluster core. According to DFT calculations, the core of each cluster species may be derived from an inner Ga₆ “cube” with two missing vertices. These are situated opposite to each other on the diagonal of the cube (see manuscript, Figure 5a). Consequently, the missing vertices are “filled up” with additional atoms (e.g. in a M₁₃ or M₁₄ cluster). This results in no significant change of neither the inner core nor the overall cluster geometry according to DFT calculations. Additionally, we have not found any evidence so far that the presented M₁₃ and M₁₄ Ni/Ga clusters may exist in drastically different structures in the solid state despite differing numbers of atoms.

“Is it possible to experimentally figure out the presence of other isomeric forms of the clusters reported here? Have the authors considered measuring ion mobility spectra of the ions? Comparing the experimental and calculated collision cross sections of the ions with the claimed structures may confirm the reported structures. Also, high resolution IMS can help in separating close lying isomeric forms.”

Based on the presented multi-analysis approach, it is not easily possible to discriminate between potential isomers. Our current setup for measuring LIFDI-MS spectra under an inert atmosphere (see Dalton Trans. 2021, 50, 26, 9031-9036, doi:10.1039/D1DT00978H) does not allow for IMS-MS experiments so far. Nevertheless, we have considered using additional advanced MS techniques in addition to LIFDI MS. However, such a setup does not exist to date to the best of our knowledge. In addition, we expect strong interference with possible degradation products if such measurements are carried out using a “standard” MS setup under atmospheric conditions due to the high sensitivity of the Ni/Ga clusters to air and moisture. Despite that, it may not be excluded that isomers close in energy exist for several (or all) clusters.

“Please explain the origin of fragment ion Ni(COD)₂ and the location of the COD ligands in the cluster structures.”

We readily observe the corresponding nickel olefin complexes (depending on the starting material) as fragment ions during LIFDI MS experiments. Most probably, they originate during ionisation from small residues of the olefin within the cluster mixtures despite washing and extensive drying *in vacuo*. Therefore, the olefin residues are most probably co-crystallised with the clusters in the solid state but they do not interact or bind strongly to them. For example, we have not observed or isolated any adduct of the corresponding olefins with clusters. Noteworthy, the intensity of the Ni(cod)₂ fragment is perceived higher in intensity than its actual concentration since the efficiency of ionization of large species like clusters and smaller molecules like metal complexes usually differ considerably.

“Please show the MS/MS data and plot the fragment ion intensity alongside the parent ions in the fragmentation related plots. There is nothing mentioned on the voltages used for the CID experiments. Also, mention the formula and m/z of the parent ions on each of the graphs.”

According to the suggestion of reviewer 1, we have added the sum formulas and m/z ratios of the corresponding clusters to the graphs of the fragmentation experiments in the supporting information. Moreover, we added the exact collision energies (in eV) to the graphs in the supporting information and two tables (Tables S23 and S24) with the relative integrals of all the detected species regarding fragmentation experiments. However, we decided to plot the corresponding fragment ion intensities separately since combining both molecular and fragment ions in one scheme would result in overcrowded figures.

“Most of the ESI figures are randomly given and not properly mentioned in the main text.”

Based on this comment, we incorporated additional references to the supporting information into the manuscript, wherever applicable.

“Minor comment: There are several typos throughout the manuscript.”

We thoroughly checked the spelling and corrected several typos in our revised manuscript.

Reviewer 2:

“... this reviewer raises one essential point on the reactive study. In this current form, it is quite difficult to follow the fate of each cluster after exposing to CO gas and silylacetylene. The authors should summarize which cluster was consumed by the MS analyses, and include one Scheme or Figure in the main text to find out the relationship of the

reactive clusters, byproduct, and unreactive clusters (as a minor revision request); otherwise, it is difficult to follow the reaction sequences.”

According to the suggestion by reviewer 2, we have added a new figure (marked as Figure 7) to the main text, including the reactivity of all described Ni/Ga clusters with CO gas to clarify the reaction pathways. In the case of (triisopropylsilyl)acetylene, one singular cluster, namely $[\text{Ni}_8\text{Ga}_6](\text{NiCp}^*)_6$, reacts to afford $[\text{Ni}_8\text{Ga}_6](\text{Cp}^*)_6(\text{TIPSA})_2$ as the only product. We also added this finding to the manuscript.

Reviewer 3:

“Roland A. Fischer and co-authors describe a rather comprehensive, multi-technology approach to investigate reaction mixtures containing polyatomic Cp-decorated Ni/Ga clusters. The class of compound has been investigated by the Fischer group for many years.”*

We thank the Reviewer for his/her work on the evaluation of our manuscript. Furthermore, we appreciate the positive feedback and the recognition of our efforts to understand at atomistic level Ni/Ga clusters decorated with Cp*.

“As outlined in their introduction, the elucidation of species being present in mixed-metal cluster mixtures is both difficult and important for learning more about the cluster growth mechanisms and about the nature of the potentially reactive species in solution. Beyond this background, the study is a nice contribution to intermetalloid cluster research.”

We thank the Reviewer for his/her positive remarks, in particular, by recognizing the importance for understanding of mixed-metal clusters in challenging environments.

“The manuscript is well written, and the results are pretty well documented, but in contrast to the statement given in the abstract, even a very thorough study of the reaction mixtures will not “overcome[s] the intrinsic limitations of poor selectivity of mixed-metal cluster syntheses and separation”. This statement is at least misleading, as the authors probably want to say that “one can overcome the lack of information” in such cases. More importantly though, this reviewer does not think that the study should be reported as a conceptually new approach for diverse reasons, and therefore requires major revision.”

We thank the Reviewer to rise up the critical issue. In light of this opinion, we agree with that comment. Thus, we revised the manuscript to avoid a particular misleading conception.

“1) The combination of HRMS studies and DFT calculation to study reaction mixtures of mixed-metal clusters (with or without ligand shell), referred to as cross-validation here, is a well-established approach, see recent works by groups working in the field of mixed-metal Zintl clusters, e.g., the groups of Sun/McGrady and Dehnen/Weigend.”

Yes, we agree with the Reviewer that our approach is not entirely novel or unique, however, the goal of our manuscript is not to demonstrate the power of the combination of several techniques for the first time. Instead of that, our manuscript has the goal to provide a deep atomistic understanding of the structure and electronic effects that leads to the stability of (Ni/Ga)Cp* clusters, which can open the room for the designing of complexes for catalytic applications.

“2) Following the description of the computational work, we consider the method not suitable to scan the entire structural space; the statement starting with “Our computational ab-initio approach, screening the complete structural space... reveals that the most stable cluster isomers are all built upon...” is thus not yet proven to apply to our understanding.”

Here, we have to clarify few points: (i) Our computational framework is not designed to scan for the entire structure space following the same strategy as global optimization techniques such as basin-hopping Monte Carlo (BHMC), please, see our paper on BHMC (<https://pubs.acs.org/doi/abs/10.1021/ci400224z>) applied to the study of finite size particles. (ii) Our computational framework is based on the two strategies: (ii.1) Based on the experimental structure obtained for (NiGa)Cp*, we performed several DFT calculations to verify and confirm the correct distribution of the Ni and Ga atoms within the NiGa structure frame determined by experimental techniques. No change is performed on the Cp* ligands. (ii.2) At a second level, we had the goal to confirm the (NiGa)Cp* structure, which is based on structure-design principles. Basically, for a given NiGa cluster, we selected a set of Ni Cp* ligands, which are placed around the NiGa cluster at random positions without overlap between the atomic species. Then, DFT calculations are performed with the aim to generate a family of structures, which can help us to improve our atomistic understanding of the electronic descriptors that drive the stability of those clusters. At the end, the lowest energy configuration of

structure family confirmed the experimental results. Therefore, our approach, which was not designed to explore all configuration space, is able to reproduce the experimental structure. (iii) Thus, to avoid the misleading of the present sentence, the manuscript was revised to provide our correct view on the computational DFT framework employed in this work.

“3) It also remains questionable whether the computational screening method is suited to generally detect the global minimum structure, especially for any system beyond the reported, very specific, Cp/Ni/Ga clusters. The closing remark “We anticipate the discovery of [...] and so far unknown, and larger intermetallic clusters and reactive superatom complexes” is thus far-fetched.”*

We would like to point out that the present approach is not a global optimization algorithm to search for the lowest energy configuration for a given set of atoms. For this particular problem, our strategy has the goal to explore the structure space of Cp* ligands around a given cluster structure. For example, in our approach, the cluster structure changes only the structure optimization, and hence, different cluster configurations must be employed to improve the search. At the end, we have a family of structures, which is employed to understand the experimental data. Furthermore, it is important to recognize that the global minimum configurations is not the most important one as those complexes are in solution and changing from one configuration to another. Thus, the identification of a family of structure to represent possible different environments is more important than identify the global minimum by itself.

“An alternative approach that does not rely on the knowledge of a crystal structure, makes use of the combination of a (very quick) calculation using first order perturbation theory in the nuclear charge and a genetic algorithm study. We assume that the methodology reported here is more complicated (thus, more time-consuming), not fully comprehensive, and not easily transferrable to other elemental combinations or unknown structures – unless demonstrated on further examples. The other method should be applied (at low costs) for comparison.”

We thank the Reviewer for his/her suggestion, however, it is not necessary in the present case as explained above. We agree with the Reviewer that different approaches can be used, which might have lower or higher computational cost than our framework, however, our goal is to obtain a reliable description of the structure and electronic properties to identify the descriptors that drives the stability similar DFT frameworks. The electronic analysis is an important part to solve our complexes corundum.

“This reviewer thinks that the novelty and generalizability of the combination of methods, and the DFT study in particular, is not as high as implied. Based on the published literature (in part cited in this manuscript), we do not consider this combination of analytical methods conceptually new, although properly applied to the specific cluster system. A more efficient computational method should be applied for comparison before claiming the advantage of the twofold study described herein.”

As mentioned above, we do not claim that our approach is novel by itself. In line with the Reviewer expectations, we have worked to improve our strategies towards a robust algorithm, which can be applied to identify the structure of complexes, however, it is not the focus of the present manuscript.

“The authors should tone down all statements that include terms like “novel” or “complete”, before the manuscript can be reconsidered for publication.”

We revised our manuscript and tune our wording to avoid any confusion or misconception. We thank the Reviewer for all his/her observations to make our scientific contribution clear.

Yours sincerely,

Prof. Jean-Yves Saillard, Prof. Juarez L. F. Da Silva & Prof. Roland A. Fischer

REVIEWERS' COMMENTS:

Reviewer #1 (Remarks to the Author):

I recommend acceptance of the article as it is in the revised form.

Reviewer #2 (Remarks to the Author):

Fischer et al. included their revision in this second manuscript, and my comments on the overall scheme for structure-activity relationship is basically responded. Still, unclarity of the sentences and inserted Figure 6 is different description of clusters in main text and Figure 6. Numbering of the clusters in Figure 6 is supportive for readers to grasp the overall reactivity (as a minor comment). After the suitable revision, this manuscript is ready for accepting to Communication Chemistry.

Reviewer #3 (Remarks to the Author):

The authors have revised their work in response to the reviewers' comments. In particular, they have toned down the generalization of their approach for all types of organometallic clusters, which is appreciated. The paper is now almost acceptable for publication. Nevertheless, I think that the sentence "Our computational ab-initio approach, screening the complete structural space of $[\text{Ni}_6/\text{Ga}_6/\text{Ga}_7](\text{Cp}^*)_6$, reveals that the most stable cluster isomers are all built upon the same structural prototype." should be changed to "Our computational ab-initio approach; investigating all possible compositional isomers $[\text{Ni}_6/\text{Ga}_6/\text{Ga}_7](\text{Cp}^*)_6$, reveals that the most stable cluster isomers are all built upon the same structural prototype." Note that "the entire structural space" would include many more structural isomers - not just those shown here. The isomers considered here were all computed based on the octahedral arrangement of $(\text{NiCp}^*)_6$ moieties as input structures and therefore exhibit a rather similar, spherical topology and therefore do not cover the entire structural space (the application of genetic algorithms would probably show this). Therefore, the statement should be modified.

As soon as this change has been made, I recommend publication in Communications Chemistry.

Communications Chemistry
Editorial Office

- electronically submitted -

Garching, December 18th, 2023

Revision for article: "**Assignment of individual structures from intermetalloid nickel gallium cluster ensembles**", Maximilian Muhr, Johannes Stephan, Lena Staiger, Karina Hemmer, Max Schütz, Patricia Heiß, Christian Jandl, Mirza Cokoja, Tim Kratky, Sebastian Günther, Dominik Huber, Samia Kahlal, Jean-Yves Saillard,* Olivier Cador, Augusto C. H. Da Silva, Juarez L. F. Da Silva,* Janos Mink, Christian Gemel, and Roland A. Fischer*

Dear Editors and Referees,

Thank you for considering our manuscript for publication as a regular article in *Communications Chemistry*. We have considered all the reviewers' critics and suggestions in this revised version of the manuscript. Below, we address every comment and provide a summary of the changes made both in the manuscript and the supporting information.

Reviewer 2:

"Fischer et al. included their revision in this second manuscript, and my comments on the overall scheme for structure-activity relationship is basically responded. Still, unclarity of the sentences and inserted Figure 6 is different description of clusters in main text and Figure 6. Numbering of the clusters in Figure 6 is supportive for readers to grasp the overall reactivity (as a minor comment). After the suitable revision, this manuscript is ready for accepting to Communication Chemistry."

According to the suggestion by reviewer 2, we added the numbering scheme to Figure 6 to clarify reaction pathways.

Reviewer 3:

"The authors have revised their work in response to the reviewers' comments. In particular, they have toned down the generalization of their approach for all types of organometallic clusters, which is appreciated. The paper is now almost acceptable for publication. Nevertheless, I think that the sentence "Our computational ab-initio approach, screening the complete structural space of [Ni₆/7Ga₆/7](Cp)₆, reveals that the most stable cluster isomers are all built upon the same structural prototype." should be changed to "Our computational ab-initio approach; investigating all possible compositional isomers [Ni₆/7Ga₆/7](Cp*)₆, reveals that the most stable cluster isomers are all built upon the same structural prototype." Note that "the entire structural space" would include many more structural isomers - not just those shown here. The isomers considered here were all computed based on the octahedral arrangement of (NiCp*)₆ moieties as input structures and therefore exhibit a rather similar, spherical topology and therefore do not cover the entire structural space (the application of genetic algorithms would probably show this). Therefore, the statement should be modified. As soon as this change has been made, I recommend publication in Communications Chemistry."*

According to this reviewer's comment, we changed the wording of this sentence in the manuscript.

Yours sincerely,

Prof. Jean-Yves Saillard, Prof. Juarez L. F. Da Silva & Prof. Roland A. Fischer